# Plasmapheresis Reduces Mycophenolic Acid Concentration: A Study of Full AUC_0–12_ in Kidney Transplant Recipients

**DOI:** 10.3390/jcm8122084

**Published:** 2019-12-01

**Authors:** Sudarat Piyasiridej, Natavudh Townamchai, Suwasin Udomkarnjananun, Somratai Vadcharavivad, Krit Pongpirul, Salin Wattanatorn, Boonchoo Sirichindakul, Yingyos Avihingsanon, Kriang Tungsanga, Somchai Eiam-Ong, Kearkiat Praditpornsilpa

**Affiliations:** 1Division of Nephrology, Department of Medicine, Faculty of Medicine, Chulalongkorn University and King Chulalongkorn Memorial Hospital, Bangkok 10330, Thailand; yam_med16@hotmail.com (S.P.); suwasin.u@gmail.com (S.U.); salin.tob@gmail.com (S.W.); yingyos.a@gmail.com (Y.A.); kriangtungsanga@hotmail.com (K.T.); somchai80754@yahoo.com (S.E.-O.); kearkiat@hotmail.com (K.P.); 2Excellence Center for Solid Organ Transplantation, King Chulalongkorn Memorial Hospital, Bangkok 10330, Thailand; boonchoog1@gmail.com; 3Renal Immunology and Renal Transplant Research Unit, Department of Medicine, Chulalongkorn University, Bangkok 10330, Thailand; 4Department of Pharmacy Practice, Faculty of Pharmaceutical Sciences, Chulalongkorn University, Bangkok 10330, Thailand; somratai.v@pharm.chula.ac.th; 5Department of Preventive and Social Medicine, Faculty of Medicine, Chulalongkorn University, Bangkok 10330, Thailand; doctorkrit@gmail.com; 6Department of International Health, Johns Hopkins Bloomberg School of Public Health, Johns Hopkins University, Baltimore, MD 21205, USA; 7Department of Surgery, Faculty of Medicine, Chulalongkorn University and King Chulalongkorn Memorial Hospital, Bangkok 10330, Thailand

**Keywords:** mycophenolic acid, immunosuppression, plasmapheresis, kidney transplantation

## Abstract

Background: Mycophenolic acid (MPA), a crucial immunosuppressive drug, and plasmapheresis, an effective immunoreduction method, are simultaneously used for the management of various immune-related diseases, including kidney transplantation. While plasmapheresis has been proven efficient in removing many substances from the blood, its effect on MPA plasma levels remains unestablished. Objectives: To evaluate the full pharmacokinetics of MPA by measuring the area under the time–concentration curve (AUC_0–12_), which is the best indicator for MPA treatment monitoring after each plasmapheresis session, and to compare the AUC_0–12_ measurements on the day with and on the day without plasmapheresis. Methods: A cross-sectional study was conducted in kidney transplantation recipients who were taking a twice-daily oral dose of mycophenolate mofetil (MMF, Cellcept^®^) and undergoing plasmapheresis at King Chulalongkorn Memorial Hospital, Bangkok, Thailand, during January 2018 and January 2019. The MPA levels were measured by an enzymatic method (Roche diagnostic^®^) 0, 1/2, 1, 2, 3, 4, 6, 8, and 12 h after MMF administration, for AUC_0–12_ calculation on the day with and on the day without plasmapheresis sessions. Plasmapheresis was started within 4 h after administering the oral morning dose of MMF. Our primary outcome was the difference of AUC_0–12_ between the day with and the day without plasmapheresis. Results: Forty complete AUC measurements included 20 measurements on the plasmapheresis day and other 20 measurements on the day without plasmapheresis in six kidney transplant patients. The mean age of the patients was 56.2 ± 20.7 years. All patients had received 1000 mg/day of MMF for at least 72 h before undergoing 3.5 ± 1.2 plasmapheresis sessions. The mean AUC on the day with plasmapheresis was lower than that on the day without plasmapheresis (28.22 ± 8.21 vs. 36.79 ± 10.29 mg × h/L, *p* = 0.001), and the percentage of AUC reduction was 19.49 ± 24.83%. This was mainly the result of a decrease in AUC_0–4_ of MPA (23.96 ± 28.12% reduction). Conclusions: Plasmapheresis significantly reduces the level of full AUC_0–12_ of MPA. The present study is the first to measure the full AUC_0–12_ in MPA-treated patients undergoing plasmapheresis. Our study suggests that a supplementary dose of MPA is necessary for patients undergoing plasmapheresis.

## 1. Introduction

Mycophenolic acid (MPA) is one of the main powerful immunosuppressive drugs widely used for many immunological diseases. There are two MPA compounds available, i.e., mycophenolate mofetil (MMF, Cellcept^®^) and enteric-coated mycophenolate sodium (EC-MPS, Myfortic^®^). Both MMF and EC-MPS are similar in terms of efficacy and safety. EC-MPS was developed to improve the side effects of upper gastrointestinal symptoms. The time to reach maximum plasma MPA concentration (t_max_) of MMF is usually within 1–2 h after an oral dose, while EC-MPS reveals a median lag time from 0.25 to 1.25 h [1]. After absorption from the gastrointestinal tract, 97 to 99% of MPA, which is the active form, will bind to serum albumin. MPA is converted by uridine diphosphate-glucuronosyltransferase (UGT) into inactive mycophenolic acid glucuronide (MPAG), which is mainly excreted by the renal tubules. MPAG can also be excreted in the biliary tract by multidrug-resistant protein (MRP), which can lead to enterohepatic recycling. [1]

Plasmapheresis is one of the most effective methods utilized for rapid immunoglobulin removal in various immunological diseases. Many proteins and protein-bound substances, including medications, can also be removed during plasmapheresis sessions [2,3]. Substances which are likely to be removed during plasmapheresis have the following characters: (1) high blood concentration, (2) high protein bound, (3) low volume of distribution (Vd), and (4) undergoing high-dose/high-efficiency plasmapheresis [4].

Several immunologically mediated diseases can be treated by MPA together with plasmapheresis, i.e., systemic lupus erythematosus (SLE), lupus nephritis, myasthenia gravis, Guillain–Barré syndrome, psoriatic arthritis, relapsed/refractory thrombotic thrombocytopenic purpura (TTP), severe polymyositis/dermatomyositis, inflammatory bowel disease, pemphigus vulgaris, and kidney transplantation [5,6,7]. Unintentional removal of MPA may result in inadequate immunosuppression and unfavorable outcomes. Of interest, the effect of plasmapheresis on MPA concentration has been studied only in a case series of two patients, one kidney transplant recipient and one patient with myasthenia gravis [8]. MPA removal were measured by considering MPA levels at only two time points—before and after each plasmapheresis session. The MPA removal was calculated on the basis of MPA concentration in plasma effluent. The authors concluded that plasmapheresis of 3 L of plasma did not significantly alter post-plasmapheresis MPA concentration. Currently, there are no available data regarding the effect of plasmapheresis on the area under the concentration–time curve from 0 to 12 h (AUC_0–12_) of MPA, which is the best indicator of MPA exposure of patients.

The present study was conducted in kidney transplant recipients who were taking stable doses of MMF and had indication for plasmapheresis to examine the effects of plasmapheresis on MPA exposure.

## 2. Methods

An observational study of patients who were taking MMF (Roche, Basel, Switzerland) in combination with plasmapheresis treatment was conducted in King Chulalongkorn Memorial Hospital, Bangkok, Thailand, during January 2018 and January 2019. The inclusion criteria were kidney transplant recipients older than 18 years, who were under an immunosuppressive regimen of tacrolimus, MMF, low-dose prednisolone and had an indication for plasmapheresis. The dosage of MMF had to be 500 mg orally every 12 h for at least one week. Exclusion criteria were patients with serum albumin concentration lower than 2 g/dL and patients who were coadministered a proton pump inhibitor.

Plasmapheresis sessions were initiated within 4 h after the morning dose of MMF. The plasmapheresis machine was Plasauto EZ^®^, and the dialyzer was Plasmaflo^®^ with a maximum pore size of 0.3 µm. The total treatment volume was 1.5 plasma volume per session. The blood flow rate was 150 mL/h. The replacement fluid was 5% albumin in the same volume as the treatment volume. The number of sessions required was determined on the basis of the clinical judgment of the attending nephrologists.

Plasmapheresis was performed on an alternate day basis for patients who were prescribed more than one plasmapheresis session.

Patients had to strictly take a stable dose of MMF, i.e., 500 mg orally every 12 h for at least one week, before entering the study. MMF dosage adjustment was not allowed during the study period. Patients were not allowed to have a meal for one hour before and two hours after taking the MMF dose. MPA level was measured by an enzymatic immunoassay method (Roche-diagnostic^®^). The AUC_0–12_ was calculated with the trapezoidal rule from the MPA levels at nine time points after the morning dose of MMF (C0, C0.5, C1, C2, C3, C4, C6, C8, and C12) (Figure 1). The full AUC_0–12_ was measured on the day just before the day patients underwent plasmapheresis and compared with the AUC_0–12_ of the following day, in which patients received the plasmapheresis treatment. Blood samples were taken via a heparin lock in the arm by using the double-syringe technique.

A complete clinical evaluation including vital signs and body weight was performed. The baseline characteristics including age, cause of end-stage renal disease, type of kidney transplantation, time after kidney transplantation, renal function, indications for plasmapheresis, session of plasmapheresis, and plasma volume per session were recorded.

Absolute and relative frequencies were used for qualitative data. Mean and standard deviation were utilized for numerical data. The chi-squared test was used for comparisons between categorical data. Paired-samples *t*-test was used to compare the AUC_0–12_ of the day with plasmapheresis and the AUC_0–12_ of the day without plasmapheresis. Data were analyzed using the SPSS statistic version 22 (IBM; New York, NY, USA).

This study was approved by The Research Ethics Review Committee for Research Involving Human Research Participants, Health Sciences Group, Chulalongkorn University (IRB No.CF 333/61). The study was registered with the Thai Clinical Trials Registry (TCTR20190211001).

## 3. Results

Six kidney transplant recipients were enrolled, with a total of 20 plasmapheresis sessions. There were 40 AUC_0–12_ measurements (each AUC consisted of measurements of MPA levels at 9 time points), 20 of which were recorded on the day just before the day patients underwent plasmapheresis, and the other 20 were recorded on the following day, when patients underwent a plasmapheresis session. The mean (±SD) age of the patients was 56.2 ± 20.7 years, and five patients (83.3%) were men (Table 1). At baseline, the mean (±SD) estimated glomerular filtration rate (eGFR) was 49.7 ± 10.9 mL/min/1.73 m^2^, serum albumin concentration was 3.8 ± 0.4 g/dL, and hemoglobin concentration was 10.3 ± 1.4 g/dL. Indication for plasmapheresis was antibody-mediated rejection (ABMR) for all six patients, who were diagnosed by pathological presentation and donor-specific antibody (DSA) detection. The number of plasmapheresis sessions per patient was 3.5 ± 1.2 (range of 1–4 sessions).

The mean of MPA AUC_0–12_ of the day with plasmapheresis was significantly lower than that of the day without plasmapheresis (28.22 ± 8.21 vs. 36.79 ± 10.29 mg × h/L, *p* = 0.001) (Figure 2). The percentage reduction of AUC_0–12_ was 19.49 ± 24.83% (Table 2). The early part of the AUC was affected by plasmapheresis sessions. The AUC_0–4_ of the day with plasmapheresis was significantly lower than that of the day without plasmapheresis (15.79 ± 6.46 vs. 21.78 ± 5.66 mg × h/L, *p* < 0.001), while the AUC_4–12_ was not significantly different between the day with and that without plasmapheresis (12.43 ± 5.02 vs. 15.00 ± 7.56 mg × h/L, *p* = 0.125).

The reduction of MPA AUC_0–12_ was detected as early as the first session of plasmapheresis. The MPA AUC_0–12_ of the day before and of the day of the first session of plasmapheresis were 41.66 ± 10.66 and 32.26 ± 9.42mg × h/L, respectively (*p* = 0.001) (Table 2 and Figure 3). The percentage reduction of MPA AUC_0–12_ of the first day of plasmapheresis session was 22.86 ± 6.99%. The AUC_0–12_ of the day before the second to that of the day of the forth plasmapheresis sessions could be rebounded from the AUC_0–12_ of the day with plasmapheresis. However, the rebounded AUC_0–12_ gradually decreased with the number of sessions of plasmapheresis that the patients received (Figure 4). Given that the target therapeutic AUC_0–12_ of MPA is 30 to 60 mg × h/L for kidney transplantation recipients [9], 17 out of 20 (85%) AUC_0–12_ measured on the day without plasmapheresis achieved the target therapeutic range, compared with only 9 out of 20 (45%) AUC_0–12_ measured on the day with plasmapheresis (*p* = 0.008) (Figure 5).

## 4. Discussion

The present study is the first to demonstrate the effect of plasmapheresis on MPA exposure by using the full MPA AUC_0–12_. The AUC_0–12_ of MPA was significantly affected by plasmapheresis. This effect was found starting from the first session of plasmapheresis (Figure 2 and Figure 3). One-fifth of the total AUC_0–12_ was lowered by plasmapheresis. The component of AUC most affected by plasmapheresis was the early part (AUC_0–4_). Undergoing plasmapheresis treatment immediately after an oral dose of MMF can lower the MPA peak level, leading to exposure to a subtherapeutic level of MPA. Consecutive sessions of plasmapheresis could increase the risk of underimmunosuppression by lowering the rebound of MPA AUC_0–12_ (Figure 4).

MMF is one of the major immunosuppressive agents widely used to treat many immunological diseases. Since overimmunosuppression can lead to many side effects and underimmunosuppression can cause unfavorable treatment outcomes, MPA level monitoring has been recommended to maintain MPA concentration at the therapeutic level [9,10]. Plasmapheresis is one of the most effective methods for rapid immunoglobulin G (IgG) reduction [5]. Many high-molecular-weight substances can also be removed during a plasmapheresis session, especially proteins and albumin, which makes albumin replacement necessary. Since 97 to 99% of MPA is protein-bound, MPA should be theoretically removed from patients during plasmapheresis treatment.

The effect of plasmapheresis on MPA plasma level was reported in only two patients who were administered MMF in combination with plasmapheresis [8]. Plasmapheresis sessions were started 4 h after MMF administration, and MPA removal was assessed at only two time points (pre- and post-plasmapheresis) together with MPA concentration in plasma waste. The authors concluded that a plasmapheresis session starting later than 4 h after the administration of an oral MMF dose did not significantly alter MPA concentration. Since serum proteins can be trapped in the dialyzer and bloodline, monitoring of MPA removal by only measuring MPA in plasma waste may not reflect total MPA removal. Our study monitored MPA exposure by full AUC_0–12_ measurement on the day with a plasmapheresis session as the study arm and on the day without plasmapheresis as the control arm. Alteration in AUC_0–12_ between the day with and the day without plasmapheresis is the best indicator of the effect of plasmapheresis on MPA plasma levels. The early phase of the full MPA AUC (peak level, AUC_0–4_) is the one mostly affecting MPA exposure and represents more than 50% of AUC_0–12_. The plasmapheresis sessions designed in the present study started within one hour after oral administration of an MMF dose which is the most crucial period for determining the effects of plasmapheresis on MPA.

MPA together with plasmapheresis is mainly utilized for the treatment of many immunologic conditions and diseases which require potent immunosuppression, such as kidney transplant rejection, severe lupus nephritis, or relapsed/refractory thrombotic thrombocytopenic purpura. The patients enrolled in the present study were kidney transplant recipients who were taking MMF and experienced antibody-mediated rejection, which is indicated for plasmapheresis treatment. The present study reveals that MPA administration without dosage adjustment during consecutive sessions of plasmapheresis can lead to unexpected underimmunosuppression and may increase the failure rate of treatment. The present study demonstrated that MPA AUC_0–12_ is reduced by 20% when a plasmapheresis session is started within 4 h after oral administration of MMF (Table 2, Figure 2). The higher the number of consecutive sessions of plasmapheresis performed, the higher the chance of MPA underexposure (Figure 4). We also further examined the role of MMF dose increments in two patients who underwent plasmapheresis and found that increasing the MMF dose from 1000 mg/day to 1250 mg/day can prevent subtherapeutic AUC_0–12_ during plasmapheresis sessions (unpublished data). An MMF dosage increment of 20% may be required to maintain a therapeutic level of MPA on the day patients undergo plasmapheresis. A further comprehensive study of therapeutic drug monitoring in patients with increased dose of MPA before undergoing plasmapheresis is crucially required. Otherwise, a 4 h delay of the plasmapheresis session after administration of an MMF dose may reduce the effect of plasmapheresis on MPA exposure (Figure 6).

The MMF dose used in the present study was relatively low. This is because the target population of patients enrolled in this study were kidney transplant recipients who were in the maintenance phase of immunosuppression. Moreover, a study on Asian patients showed that most of the patients achieved the target MPA level with an MMF dose of 1000 mg/day [11]. Besides conventional plasmapheresis, a study of the effects of others apheresis techniques such as double-filtration plasmapheresis and immunoadsorption, which have different kinetics of protein removal, should be carried out.

## 5. Conclusions

Plasmapheresis significantly reduces MPA plasma levels, particularly in the early phase after oral administration of an MPA dose. This effect should be addressed when combining MPA administration together with plasmapheresis in a treatment protocol.

## Figures and Tables

**Figure 1 jcm-08-02084-f001:**
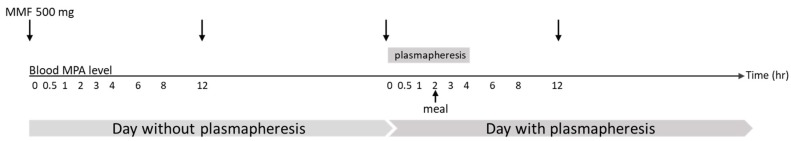
Timing of mofetil (MMF) dosage, plasmapheresis sessions, and meal on the day before and on the day with a plasmapheresis session. MPA: mycophenolic acid.

**Figure 2 jcm-08-02084-f002:**
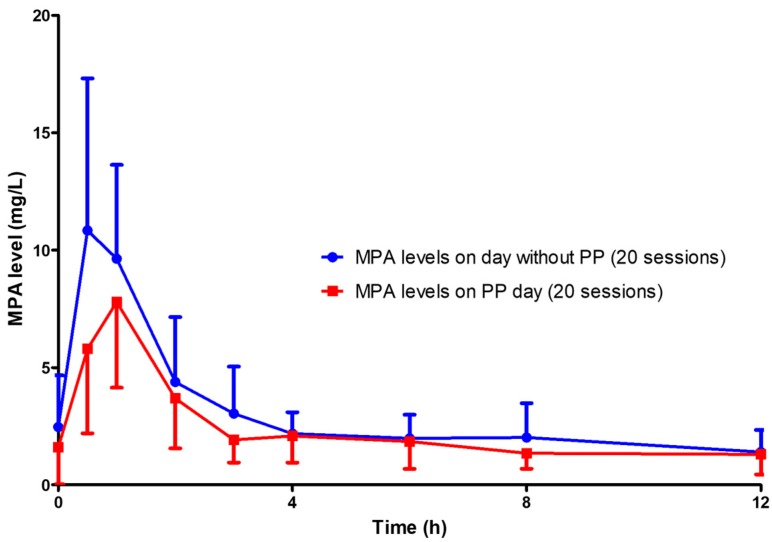
MPA levels on the day with plasmapheresis (20 sessions) compared with those on the day without plasmapheresis (20 sessions). PP: plasmapheresis.

**Figure 3 jcm-08-02084-f003:**
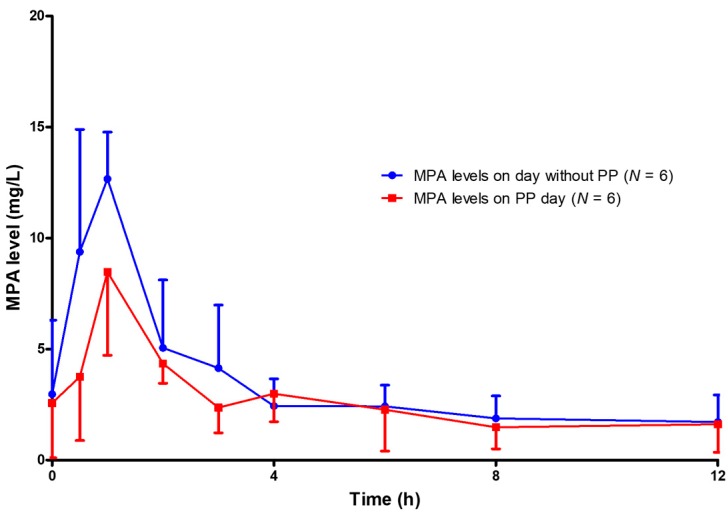
MPA levels on the day before the first plasmapheresis session (*N* = 6) compared with MPA levels on the day with the first plasmapheresis session (*N* = 6).

**Figure 4 jcm-08-02084-f004:**
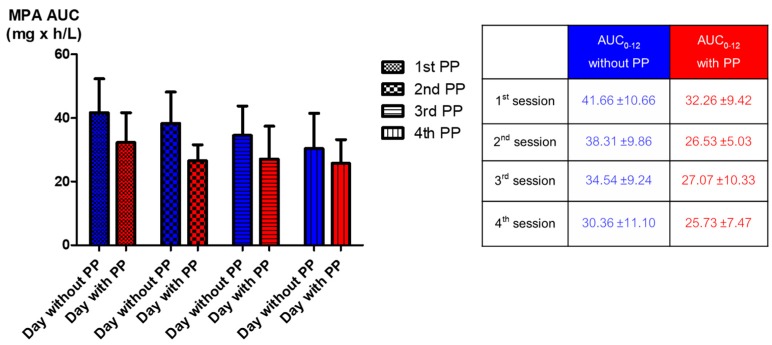
Comparison of the mean MPA AUC_0–12_ between the day with and that without plasmapheresis from the first plasmapheresis session to the fourth session.

**Figure 5 jcm-08-02084-f005:**
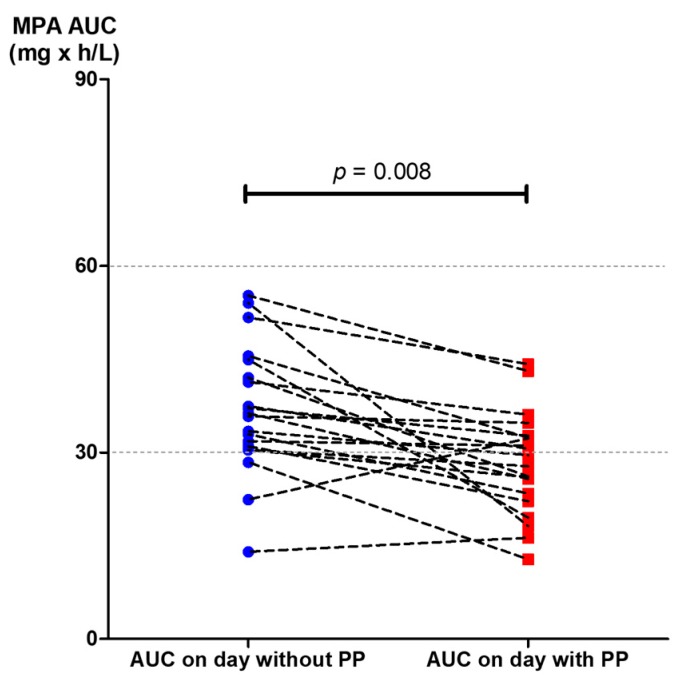
The MPA AUC_0–12_ achieved the target level between the day just before a plasmapheresis session (20 measurements) and the following day, when plasmapheresis was administered (20 measurements).

**Figure 6 jcm-08-02084-f006:**
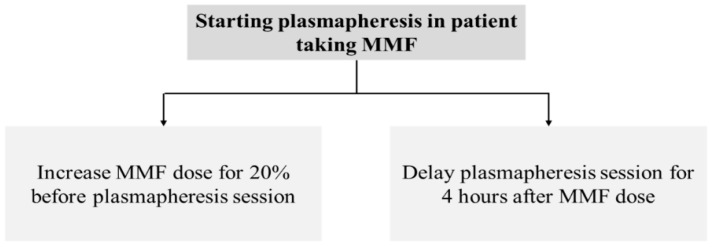
Recommendations for MMF dose or plasmapheresis adjustment in patient receiving concomitant MMF and plasmapheresis treatment.

**Table 1 jcm-08-02084-t001:** Baseline characteristics of the patients.

Characteristics	
Age, year (mean ± SD) (range)	56.2 ± 20.7 (25–80)
Male (*n*, %)	5/6, 83%
Cause of ESRD before kidney transplantation	
Unknown (*n*, %)	6/6, 100%
Type of kidney transplantation Living donor kidney transplantation (*n*, %)	2/6, 33%
History of previous kidney transplantation (*n*, %)	1/6, 16.7%
HLA mismatch (*n*, %)	
0	0/6
1–5	6/6, 100%
6	0/6
Panel reactive antibody (*n*, %)	
0%	4/6, 66.7%
1–80%	0/6
More than 80%	2/6, 33.3%
Induction immunosuppression (*n*, %)	
Anti-IL2 receptor antibody	4/6, 66.7%
Anti-thymocyte globulin	2/6, 33.3%
Time after transplantation, month (mean ± SD) (range)	97.1 ± 69.5 (1.97–196.52)
Body weight, kg (mean ± SD) (range)	62.2 ± 12.4 (42.7–79.3)
eGFR CKD-EPI, mL/min/1.73 m ^2^ (mean ± SD)	49.7 ± 10.9
Serum albumin, mg/dL (mean ± SD) (range)	3.8 ± 0.4 (3.0–4.2)
Hemoglobin, mg/dL (mean ± SD) (range)	10.3 ± 1.4 9.0–12.2
Liver enzyme, U/L (mean ± SD) SGOT (range) SGPT (range)	32 ± 42 (10–117) 33 ± 36 (10–104)
Type of plasmapheresis	
Conventional plasmapheresis (*n*, %)	6/6, 100%
Indication for plasmapheresis (*n*, %)	
ABMR	6/6, 100%
Acute ABMR	2/6, 33.3%
Chronic active ABMR	4/6, 66.7%
Plasma volume per session, mL (mean ± SD)	4,041 ± 749
Number of plasmapheresis session in each patient (mean ± SD)	3.5 ± 1.2

ESRD: end-stage renal disease, ABMR: antibody-mediated rejection; eGFR: estimated glomerular filtration rate; CKD-EPI: chronic kidney disease epidemiology collaboration; SGOT: serum glutamic-oxaloacetic transaminase; SGPT: serum glutamate-pyruvate transaminase.

**Table 2 jcm-08-02084-t002:** Comparison of MPA AUCs recorded on days with and without plasmapheresis, from 0 to 12 h, from 0 to 4 h, and from 4 to 12 h.

Parameters	Day without Plasmapheresis	Day with Plasmapheresis	*p*-Value
AUC_0–12_ mg × h/L (mean ± SD)	36.79 ± 10.29	28.22 ± 8.21	*p* = 0.001
Percentage reduction of AUC_0–12_ (%)	19.49 ± 24.83		-
AUC_0–4_ mg × h/L (mean ± SD)	21.78 ± 5.66	15.79 ± 6.46	*p* < 0.001
Percentage reduction of AUC_0–4_ (%)	23.96 ± 28.12		-
AUC_4–12_ mg × h/L (mean ± SD)	15.00 ± 7.56	12.43 ± 5.02	*p* = 0.125
Percentage reduction of AUC_4–12_ (%)	3.88 ± 42.89		-
AUC_0–12_ of the first day with plasmapheresis session, mg × h/L (mean ± SD)	41.66 ± 10.66	32.26 ± 9.42	*p* = 0.001
Percentage reduction of AUC_0–12_ of the first day with plasmapheresis session (%)	22.86 ± 6.99		-

(AUC; area under the time–concentration curve).

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
