# Peer review of "Plasmapheresis Reduces Mycophenolic Acid Concentration: A Study of Full AUC0–12 in Kidney Transplant Recipients"

_jcm, 2019, doi:10.3390/jcm8122084_

Round 1
Reviewer 1 Report
The article by Piyasiridej S et al. is on an interesting clinical study on the effect of plasmapheresis on MPA levels in patients with ESRF.
However, there are some issues in the manuscript that needs to be addressed to provide clarity to the information presented. Study design and results section are somewhat confusing because sufficient information has not been provided for the reader to understand the study and its outcome. Results explanation and interpretation are poor. Recommend to re-write these two sections providing details of the study design and clear explanation of the results. Please see the detailed comments under each section.
Title: Recommended to include the information that the study was on ESRF patients. Title is too general, needs to be specific because the authors are discussing about a specific group of patients.
Introduction
What is/are the difference/s between two compounds mycophenolate mofetil and mycophenolate sodium?
Line 59: ‘rapidly’ needs to be replaced with ‘rapid’
Line 61: Add ‘to be’ after likely
Line 71: Does this case series study on renal transplant patients? Check the article “Changes in blood concentration of mycophenolic acid and FK506 in a heart-transplant patient treated with plasmapheresis” Int J Clin Pharmacol Ther. 2019 Jan;57(1):32-36
Lines 71-73: The sentence ‘This case series…’ needs to be rewritten to provide a better read
Line 79: MPA exposure or concentration?
Methods
Line 88: ‘hours’ not ‘hour’
Line 88: What is the dose of MMF provided?
Lines 93-95: ‘MMF 500mg bd for at least 1 week’, this needs to be included as an inclusion criteria. Then, the authors don’t have to mention the dose in Line 88.
Did all the participants had this dose during the study, while having plasmapheresis and the day without plasmapheresis?
Did the researchers collect the samples for the day without plasmapheresis immediately after completing the sample collection for the day with plasmapheresis? Was there a time gap between the two sample collection phases? If so, were the participants given MMF 500mg bd during this time gap?
Line 95: Are you referring to ‘enzymatic immunoassay’?
There’s no information on blood collection.
Line 97: What does C0 refers to? Is this the time the patients had the morning dose of MMF? At what time point/s did they have the plasmapheresis? Is this similar to all 6 patients or did they have plasmapheresis at different times. This is confusing, authors should explain this clearly. Or provide a schematic diagram to indicate plasmapheresis time points and sample collection time points
Line 99: Repetition of information (plasmapheresis sessions were…..) Same as line 88
Line 100:Should be ‘signs’ not ‘sign’
Line 102: What do you mean by ‘cause of plasmapheresis’?
Line 105: Do you mean ‘numerical data’?
What statistical tests were used to compare the MPA levels (with and without plasmapheresis)
Results
It seems that authors are using the ‘mean’ MPA values to compare, mention this when explaining the results. Authors have mentioned this in the Table but not in the text when explaining the results
Throughout the article the authors mention ‘AUC0-12’ even when referring to a single time point or even in axis labelling in the graphs (e.g. Fig 3) This is not correct. Use the information of the specific time points when referring to AUC or when using in general, use just ‘AUC’ when explaining. Otherwise this is very confusing to the reader.
Page 3, Line 120: ‘reduction ratio’, are you referring to percentage reduction. If so, this needs to be clearly mentioned here and in Table 2. If not, explain how the reduction ratio was calculated
Line 123-124: Do you mean not ‘significantly’ different? You cannot say it is not different because there is some difference in the provided mean values
Table 1: Include the minimum and maximum levels for quantitative data
Figure 1: x axis label-concentration time is not a suitable axis title. ‘n’ should be 6, not 20
Fig 1 and Fig 2: Figure presents the MPA concentration against the time points, not AUC. However, legend of the graph mentions about AUC, explain.
What is the difference between fig 1 and 2? This is confusing. There were six patients and it is not clear how they ended up with a total of 40 measurements. You need to provide some information to explain this. There are 9 time points and two different phases of the study. Based on this information, authors have not collected samples at each time point for every patient. This can affect the quality of evidence provided.
Table 2: Provide AUC 4-12 Reduction (%) for the consistency of the table
Table 2: Provide the unit for AUC measurement
Table 2: What do you mean by ‘First AUC0-12?
Lines 138-140: What is the statistical test used to compare this data?
Fig 4: This is very confusing. How did you do this comparison? It seems that authors have used 20 pairs for this comparison. Are these from the same time points? It is mentioned that the number of plasmapheresis sessions for each patient was not similar (number of sessions 3.5±1.2). How did the authors address the effect of the number and the timing of plasmapheresis on MPA concentration for this comparison?
Discussion
This study has been limited to ESRF patients whose cause of ESRF is not known. There is no discussion or explanation why the study population was limited to this group.
What is the purpose of figure 5?This information can easily and clearly be presented in the main text. The figure does not add any additional information.
Author Response
We thank you for your excellent and constructive comments. Here are our point-by-point responses below:
Title: Recommended to include the information that the study was on ESRF patients. Title is too general, needs to be specific because the authors are discussing about a specific group of patients.
Reply: The population of Kidney Transplant Recipients has been added to the title
Introduction
What is/are the difference/s between two compounds mycophenolate mofetil and mycophenolate sodium?
Reply: The differences between these two drugs were added
Line 59: ‘rapidly’ needs to be replaced with ‘rapid’
Reply: Change from rapidly to rapid
Line 61: Add ‘to be’ after likely
Reply: “to be” was added
Line 71: Does this case series study on renal transplant patients? Check the article “Changes in blood concentration of mycophenolic acid and FK506 in a heart-transplant patient treated with plasmapheresis” Int J Clin Pharmacol Ther. 2019 Jan;57(1):32-36
Reply: This study has been done in kidney transplant recipients. The blood tacrolimus (FK506) normally was not affected by plasmapheresis, because more than 90% of tacrolimus distribute in red blood cell which cannot be removed by plasmapheresis.
Lines 71-73: The sentence ‘This case series…’ needs to be rewritten to provide a better read
Reply: This sentence has been rewritten
Line 79: MPA exposure or concentration?
Reply: The MPA exposure, which means the overall MPA concentration patient received
Methods
Line 88: ‘hours’ not ‘hour’
Reply: Changed to hour
Line 88: What is the dose of MMF provided?
Reply: The dose of MMF 500 mg orally every 12 hours was added in inclusion criteria
Lines 93-95: ‘MMF 500mg bd for at least 1 week’, this needs to be included as an inclusion criteria. Then, the authors don’t have to mention the dose in Line 88.
Reply: The dose of MMF 500 mg orally every 12 hours was added in inclusion criteria
Did all the participants had this dose during the study, while having plasmapheresis and the day without plasmapheresis?
Reply: All patients must receive MMF 500 mg every 12 hours every day during study period.
Did the researchers collect the samples for the day without plasmapheresis immediately after completing the sample collection for the day with plasmapheresis? Was there a time gap between the two sample collection phases? If so, were the participants given MMF 500mg bd during this time gap?
Reply: Since plasmapheresis has been performed in the alternate day basis, there was no gap between blood collection on the day without plasmapheresis and the day with plasmapheresis.
Line 95: Are you referring to ‘enzymatic immunoassay’?
Reply: Enzymatic immunoassay was added
There’s no information on blood collection.
Reply: Details about blood collection was added
Line 97: What does C0 refers to? Is this the time the patients had the morning dose of MMF? At what time point/s did they have the plasmapheresis? Is this similar to all 6 patients or did they have plasmapheresis at different times. This is confusing, authors should explain this clearly. Or provide a schematic diagram to indicate plasmapheresis time points and sample collection time points
Reply: Figure 1 was added to explain protocol and time point of blood collection, meal, and plasmapheresis.
Line 99: Repetition of information (plasmapheresis sessions were…..) Same as line 88
Reply: The repetition sentence was removed
Line 100: Should be ‘signs’ not ‘sign’
Reply: Changed to sign
Line 102: What do you mean by ‘cause of plasmapheresis’?
Reply: Changed to indication for plasmapheresis
Line 105: Do you mean ‘numerical data’?
Reply: Change to numerical data
What statistical tests were used to compare the MPA levels (with and without plasmapheresis)
Reply: The additional statistic used was added
Results
It seems that authors are using the ‘mean’ MPA values to compare, mention this when explaining the results. Authors have mentioned this in the Table but not in the text when explaining the results
Throughout the article the authors mention ‘AUC0-12’ even when referring to a single time point or even in axis labelling in the graphs (e.g. Fig 3) This is not correct. Use the information of the specific time points when referring to AUC or when using in general, use just ‘AUC’ when explaining. Otherwise this is very confusing to the reader.
Reply: Figures (axis labelling) have been corrected. Some inappropriate AUC0-12 were changed to MPA level.
Page 3, Line 120: ‘reduction ratio’, are you referring to percentage reduction. If so, this needs to be clearly mentioned here and in Table 2. If not, explain how the reduction ratio was calculated
Reply: Percentage reduction was used instead of reduction ratio
Line 123-124: Do you mean not ‘significantly’ different? You cannot say it is not different because there is some difference in the provided mean values
Reply: Changed to significantly different
Table 1: Include the minimum and maximum levels for quantitative data
Reply: Minimum and maximum levels of data were added to table
Figure 1: x axis label-concentration time is not a suitable axis title. ‘n’ should be 6, not 20
Reply: N=20 was changed to 20 sessions
Fig 1 and Fig 2: Figure presents the MPA concentration against the time points, not AUC. However, legend of the graph mentions about AUC, explain.
Reply: AUC was changed to MPA level
What is the difference between fig 1 and 2? This is confusing. There were six patients and it is not clear how they ended up with a total of 40 measurements. You need to provide some information to explain this. There are 9 time points and two different phases of the study. Based on this information, authors have not collected samples at each time point for every patient. This can affect the quality of evidence provided.
Reply: The explanation was added. There were 40 times of full AUC measurement in this study. Each AUC consist of 9 time points of MPA level. 20 AUCs were measured on the day just before the day of plasmapheresis and another 20 AUCs were measured on the consecutive day that has plasmapheresis session.
Table 2: Provide AUC 4-12 Reduction (%) for the consistency of the table
Reply: The percentage reduction was added
Table 2: Provide the unit for AUC measurement
Reply: The unit for AUC was added
Table 2: What do you mean by ‘First AUC0-12?
Reply: Changed to AUC of the first day with plasmapheresis session
Lines 138-140: What is the statistical test used to compare this data?
Reply: The statistics used were added
Fig 4: This is very confusing. How did you do this comparison? It seems that authors have used 20 pairs for this comparison. Are these from the same time points? It is mentioned that the number of plasmapheresis sessions for each patient was not similar (number of sessions 3.5±1.2). How did the authors address the effect of the number and the timing of plasmapheresis on MPA concentration for this comparison?
Reply: Figure 4 (now becomes figure 5) the detail of comparison between the day without plasmapheresis with the consecutive day with plasmapheresis was added in part of method
Discussion
This study has been limited to ESRF patients whose cause of ESRF is not known. There is no discussion or explanation why the study population was limited to this group.
Reply: Kidney transplant recipient with MMF and rejection episode which require plasmapheresis treatment was added in discussion.
What is the purpose of figure 5? This information can easily and clearly be presented in the main text. The figure does not add any additional information.
Reply: We hope that the figure 5 (now become figure 6) can be easier to understand.
Sincerely,
Reviewer 2 Report
This is an interesting study on pharmacokinetics of MMF during PE. The results are very interesting since both MMF and Pes are an essential part of therapy for many immunological disorders. Although, the study was executed quite well from a pharmacological point of few and the results are clear, some information about the clinical circumstances are missing and should be addressed.
When was the humoral rejection observed? In the early or in a later phase? Was it a purely humoral or mixed humoral and cellular? Which was the immunological risk of the recipient? (PRAs, DSAs, number of Transplantations, HLA-MM). Which was the induction therapy? Where there any complications related to PE? Was the substitution fluid only albumin or where there also sessions done with FFPs or mixed FFPs and HA as often occurs in an acute clinical setting, such as humoral rejection? Which was the further follow up of the patients? Was the PE as usually conducted after the results of fibrinogen levels were available? Which was the cut-off to decide if the next PE should be done or not? The authors had the opportunity to measure Tacrolimus levels too and must simply explain why this was not done.
Author Response
We thank you for your suggestions. Here are our point-by-point responses below:
When was the humoral rejection observed? In the early or in a later phase? Was it a purely humoral or mixed humoral and cellular? Which was the immunological risk of the recipient? (PRAs, DSAs, number of Transplantations, HLA-MM). Which was the induction therapy? Where there any complications related to PE? Was the substitution fluid only albumin or where there also sessions done with FFPs or mixed FFPs and HA as often occurs in an acute clinical setting, such as humoral rejection? Which was the further follow up of the patients? Was the PE as usually conducted after the results of fibrinogen levels were available? Which was the cut-off to decide if the next PE should be done or not? The authors had the opportunity to measure Tacrolimus levels too and must simply explain why this was not done
Reply: All clinical data are added i.e. time after transplantation to plasmapheresis treatment, type of rejection, baseline immunologic risk of all 6 patients, induction treatment, replacement fluid. Number of plasmapheresis sessions depend on clinical and MFI of DSA and has been decided by physician. The tacrolimus level was measured for only clinical / treatment purpose. Tacrolimus mainly (more than 90%) distribute in red blood cell which not affected by plasmapheresis. It is different from MMF which has albumin bound for more than 90% and should be easily removed from blood by plasmapheresis. We used to collect tacrolimus level for our patient during plasmapheresis before this study and found that plasmapheresis has no effect on tacrolimus concentration. We would like to mainly focus on MMF which data on effect of plasmapheresis remain inconclusive.
Round 2
Reviewer 1 Report
Authors have clearly explained most of the content which was not clear in their first draft. Few more comments on the revised version.
Introduction
Line 75-77: This sentence has errors in structure, please check.
Line 75: ‘case series of two patients’ be specific, if they are renal transplant patients, mention that here.
Methods
Line 100: grammar, should be past tense
Figure 1: It would be better to remove MPA level line from the figure. Just indicate plasmaparesis time MMF administering times, meal time and sample collection times. This should not include the data/results (MPA level). You may consider a different schematic diagram (e.g. flow chart) rather than a graph where you can even show the two consecutive days of sample collection to provide a clear methodology.
Results
Line 131-132: Remove ‘have been performed’
Line 134: replace which with ‘when’
Figure 3: What is the purpose of comparing the levels only related to the first day? Fig 3 provides the data for all 40 cycles of the 6 patients. What is the purpose of this additional graph limited to first day of PP? Explain.
Figure 4: Is this ‘mean’ MPA level?
Figure 5: Mention this is for the total 40 cycles or 20 with pp and 20 without pp.
Discussion
This study sample is limited to ESRF patients without a known cause (unknown aetiology). Is there any reason for this limitation (not including any patients with common causes sch as DM and HT)? It would be better if authors discuss this? Will this have any implications on their recommendation if the reason for ESRF is different (i.e. can we expect the same changes in a diabetic patient?)
Author Response
We thank you for your comments and helping correction this manuscript in this second round. Here are our point-by-point responses below:
Introduction
Line 75-77: This sentence has errors in structure, please check.
Reply: This sentence is rewritten
Line 75: ‘case series of two patients’ be specific, if they are renal transplant patients, mention that here.
Reply: one patient was myasthenia gravis and one kidney transplant have been added to sentence
Methods
Line 100: grammar, should be past tense
Reply: Corrected to past tense
Figure 1: It would be better to remove MPA level line from the figure. Just indicate plasmaparesis time MMF administering times, meal time and sample collection times. This should not include the data/results (MPA level). You may consider a different schematic diagram (e.g. flow chart) rather than a graph where you can even show the two consecutive days of sample collection to provide a clear methodology.
Reply: Figure was changed according to your recommendation
Results
Line 131-132: Remove ‘have been performed’
Reply: Have been performed was removed
Line 134: replace which with ‘when’
Reply: When was used
Figure 3: What is the purpose of comparing the levels only related to the first day? Fig 3 provides the data for all 40 cycles of the 6 patients. What is the purpose of this additional graph limited to first day of PP? Explain.
Reply: We would like to show that the effect plasmapheresis has been shown as early as the first session. This effect doesn’t need repeated sessions of plasmapheresis to be found.
Figure 4: Is this ‘mean’ MPA level?
Reply: “Mean” was added
Figure 5: Mention this is for the total 40 cycles or 20 with pp and 20 without pp.
Reply: “20 measurement” was added
Discussion
This study sample is limited to ESRF patients without a known cause (unknown aetiology). Is there any reason for this limitation (not including any patients with common causes sch as DM and HT)? It would be better if authors discuss this? Will this have any implications on their recommendation if the reason for ESRF is different (i.e. can we expect the same changes in a diabetic patient?)
Reply: DM is the most common cause of ESRD in whole world. However, kidney transplant recipient usually younger than general ESRD patients. This because ESRD patients with DM usually older and have many comorbid diseases which cannot undergo kidney transplantation. The major cause of ESRD for younger patient is glomerular disease but only some patients had kidney biopsy for diagnosis before ESRD and many patients went through ESRD without definite diagnosis by kidney biopsy.
Thank you very much
Sincerely,
Natavudh Townamchai
Assistant Professor, Department of Medicine
Faculty of Medicine, Chulalongkorn University
Bangkok, Thailand
Reviewer 2 Report
All the raised issues of the first revision have been sufficiently adressed. I have no objections to the publication of the revised manuscript in the actual version.
Author Response
Thank you very much for help improving this manuscript
Sincerely,
Natavudh Townamchai
Assistant Professor, Department of Medicine
Faculty of Medicine, Chulalongkorn University
Bangkok, Thailand